# Rapid Review on the Concept of Positive Health and Its Implementation in Practice

**DOI:** 10.3390/healthcare12060671

**Published:** 2024-03-16

**Authors:** Marja van Vliet, Miriam de Kleijn, Karolien van den Brekel-Dijkstra, Tim Huijts, Sandra van Hogen-Koster, Hans Peter Jung, Machteld Huber

**Affiliations:** 1Institute for Positive Health, 3521 AL Utrecht, The Netherlands; m.dekleijn@iph.nl (M.d.K.);; 2Positive Health International, 3572 JC Utrecht, The Netherlands; k.vandenbrekel@positivehealth-international.com; 3Leidsche Rijn Julius Centre for Primary Care, Utrecht University Medical Centre, 3584 CX Utrecht, The Netherlands; 4Research Centre for Education and the Labour Market (ROA), School of Business and Economics, Maastricht University, 6211 LM Maastricht, The Netherlands; t.huijts@maastrichtuniversity.nl; 5Health Care Academy, Saxion University of Applied Sciences, 7513 AB Enschede, The Netherlands; s.koster.01@saxion.nl; 6Medical School Twente, Medical Spectrum Twente, 7512 KZ Enschede, The Netherlands; 7Afferden Centre for Primary Care, 5851 AS Afferden, The Netherlands; hpjung@hapafferden.nl

**Keywords:** positive health, health, rapid review, shared decision-making, health concept, transition, sustainable healthcare

## Abstract

Positive health (PH) has been described as a promising transformative innovation to address the challenges of promoting well-being and reducing the burden of disease. For this study, we conducted a scientific literature review of the current state of knowledge about PH as introduced by Huber and colleagues, following the Cochrane Rapid Review recommendations. Three databases were searched (PubMed, Google Scholar, and CINAHL). Data were extracted and synthesised using a narrative approach. A total of 55 articles were included. The initial evaluation revealed promising results at both the individual and collective levels. However, several articles gave reason for further refinement of the conceptualisation of PH and of ways to measure the effects of PH interventions in greater detail. Professionals also expressed a desire for a more informed application and elaboration of the PH method, in various settings and populations, to increase its effectiveness in practice. The results from the rapid review highlight the transformative potential of PH in shifting from a disease-oriented to a health-oriented paradigm of healthcare. This underlines the need for continued research regarding further development of the concept and its practical method, along with the necessity for methodological innovation.

## 1. Introduction

Healthcare systems worldwide are striving for new ways of dealing with an increasing burden of chronic diseases, rising healthcare expenditures, and the challenges of an aging population [1,2]. Within healthcare, critics argue that there is too much emphasis on biomedical models that focus primarily on diagnosing and treating diseases, with limited attention paid to prevention, health promotion, and well-being [3,4,5,6]. Huber et al. (2011) recognise that this biomedical tendency is rooted in a view of ‘health as the absence of disease’. Huber introduced a more dynamic concept of health valuing resilience as ‘the ability to adapt and self-manage in the face of social, physical and emotional challenges’ [7]. This was deliberately described not as a ‘definition’, which would mean a demarcating criterion, but rather as a ‘general concept’ intended to characterise a goal to work towards, namely, that of increasing resilience, overall health, and well-being. This concept was further elaborated into positive health (PH), which includes the following six dimensions: bodily functions, mental well-being, meaningfulness, quality of life, participation, and daily functioning [8]. These dimensions were derived from responses of patients and citizens to the question of what they considered to be indicators of ‘health’ [8]. Internationally, various conceptualisations of the term ‘positive health’ have emerged, all sharing a common basis in the salutogenic approach, positive psychology, or lifestyle medicine, focusing on the resources that promote health rather than the mere absence of disease [9]. Thereby, health relates closely to concepts such as well-being and quality of life. However, the relationship, as well as differences between these concepts, still need to be studied in more detail [10]. For practical use, the method of PH—or the so-called ‘alternative dialogue’—was developed to support people in becoming more aware of their purpose in life, as well as finding intrinsic motivation and resilience to deal with life’s physical, emotional, and social challenges. The method was initially meant to be used during consultations with healthcare providers in healthcare and social domains (e.g., general practitioners, medical doctors, nurses, social workers). Soon afterwards, it was applied in conversations with peers or volunteers, and after that, other domains followed, such as education and work. The first step of this method involves applying the dialogue tool, with average scores on the six dimensions displayed within the positive health spiderweb (see Figure 1). When connecting the scores, the resulting visual spiderweb surface gives insight into people’s perceived own general health situation and the coherence between the dimensions.

The next step is to determine whether there is a desire to change, and, if so, in which of the dimensions. The third and final step is that of identifying whether the person involved would need support in making the required changes, and, if so, what this support would be. Reflection on the results from the spiderweb is aimed to help people direct their attention to those areas that are relevant to them. The focus, furthermore, is on the preferred future rather than on the things that went wrong in the past, and on increasing the responsivity of professionals to genuinely recognise the potential of the individual and meet their needs. Because support may be needed in more areas than the usual healthcare dimensions, the tool aims to promote integrated collaboration between and within the various domains (e.g., healthcare, social care, work environment), ultimately resulting in improving people’s personal well-being [10]. In general, PH provides a more holistic perspective in addition to a more traditional approach focusing on illness.

Specific versions of the dialogue tool (i.e., digital and on paper) are available for adults, adolescents, and children [11]. In addition, there is a simple version for adults with lower reading skills. Since their introduction in 2016, these dialogue tools have been widely used in The Netherlands, with more than 350,000 unique users (as of June 2023). International versions are also available in English, German, French, Spanish, Japanese, and Icelandic (and more are currently being developed) [12]. The statements in the adult version were based on the original 32 aspects derived from interviews and principal component analysis [8,13]. For practical use, they were later reformulated to a simpler language level (B1). Although there could be a certain loss of discriminatory quality, this was considered acceptable because the dialogue tool is intended to stimulate self-reflection rather than being a measurement instrument. In addition, at the request of professionals, a few ‘determinants’ were added to recognise homelessness and debt, which then led to a total of 44 statements. Recognising the importance of having a measurement instrument for research purposes alongside the dialogue tool, a tool, discriminating a selection of items of that tool, was subsequently developed for this particular purpose [14].

PH has been described as a concept of successful transformative innovation [1,15]. Integration of the concept (the broad vision on health) and method (spiderweb and alternative dialogue) take place at individual, organisational, community, regional and national levels within the Dutch healthcare system and beyond. The Dutch government considers PH a promising approach to promoting well-being and reducing the burden of disease [16,17]. Furthermore, its impact has transcended national borders, generating attention and gaining recognition on an international scale [10]. With the international experiences so far, in various countries, the concept and method seem to be universally applicable, with some country- and culture-specific needs. Along with the growing demand for implementation in practice, questions have arisen about PH’s impact, its scientifically evaluated effectiveness, and its potential applicability in other settings and domains. At the same time, practitioners and researchers have both expressed the need for a more extensive substantiation of the concept and for valid and reliable measurements of its impact [14,18]. To address these urgent questions, this rapid review aims to summarize the current state of knowledge on PH. This summary provides researchers, policymakers, practitioners, and other healthcare professionals with a comprehensive overview of key findings from the literature on PH and identifies knowledge gaps, needs, and areas for future research.

## 2. Materials and Methods

For this study, a rapid review approach was conducted according to the Cochrane Rapid Review recommendations [19]. Rapid reviews balance the time-sensitive demands of policymakers and methodological quality [20].

The current rapid review was carried out according to the following steps: Formulation of the explorative research question using the SPIDER framework (Sample, Phenomenon of Interest, Design, Evaluation, Research type) [21,22];Development of the search strategy;Specification of inclusion criteria;Abstract screening and study selection;Data extraction;Synthesis of findings.

The research questions that guided this rapid review are as follows: Main question:What is the current state of knowledge about PH as introduced by Huber et al. (2016) within the available scientific literature? [8]
In order to answer this question, the following sub-questions will be addressed:*What specific areas related to the PH concept and method have been investigated and what are the key findings from these studies?*What areas and/or topics can be derived from the literature that are important for the implementation of the PH concept and method at individual, organisational, community, regional, national and international levels? 


We conducted a comprehensive literature search using electronic databases between April and July 2023. The search strategy was to include all articles citing Huber et al. (2016) in Pubmed, Google Scholar, and CINAHL. The articles in English and Dutch had to have been published between 2016 (i.e., the introduction of PH by Huber et al. (2016)) and the 1st of July 2023 (i.e., date of the last search).

Studies were included if they met the following predefined eligibility criteria:Citing PH as conceptualised by Huber et al. (2016) [8];PH is part of or related to the research objective and/or study method;Original research (e.g., qualitative, quantitative, mixed-methods studies, reviews) in English or Dutch, published in a peer-reviewed scientific journal.

For the citations, each electronic database was searched to select all articles citing Huber et al. (2016). During the first exploration, a search strategy was used containing the search term ‘positive health’ in the title, abstract, or keywords. This was compared with search results with the term ‘positive health’ in the main text. As the first strategy appeared to result in too many missings and the second yielded too many irrelevant records, an alternative strategy was developed. For this strategy, we used a special function in the databases to display all articles that cited Huber et al. (2016). Next, all records were split into two batches. M.d.K. and M.v.V. independently screened the titles and abstracts of the articles retrieved within one of the batches to exclude the studies that were non-peer-reviewed publications and/or those in other languages. M.d.K. and M.v.V. performed a member check for the first 50 records of each batch, which yielded no disparities. The full text of each article was then reviewed to check whether PH was part of or related to the research objective and/or study method. Thus, articles in which PH was mentioned only in an introduction or discussion section to provide contextual information to an article were excluded. M.d.K. and M.v.V. independently reviewed all full texts of the articles. Any discord between them was resolved through a consensus meeting in which a third reviewer (K.v.d.B-D.) was consulted. 

The above search strategy resulted in 639 hits. After removing duplicates, 528 articles remained for screening and further review. Ultimately, 55 articles were finally included. See Figure 2 for the PRISMA flow diagram. 

General information (authors, year, language, study design, study population and setting) was organised in a data extraction form by M.v.V. The included articles underwent manifest content analysis, which involved a systematic process of categorising and analysing content for patterns [23]. Themes related to the research questions (see above) were identified in an iterative process of inductive coding. M.v.V. coded the texts line by line according to the content. Next, M.v.V. identified descriptive themes and sub-themes by grouping the codes according to similarities and differences and in relation to the research questions. The coding framework and emerging themes and sub-themes were iteratively discussed and refined during regular meetings in which all authors participated. A consensus was reached through discussion between all authors, resulting in iterative refinement to increase the reliability and validity of our review results.

Below, the findings from our review are discussed and summarised in a narrative format.

## 3. Results

### 3.1. Search Results

A total of 55 articles were included in the review (see Figure 2); the majority of those was written in English (73%) and published between 2020 and 2023 (65%). Nearly half concerned qualitative studies (40%), followed by quantitative studies (24%) and mixed-method studies (22%), reflections (9%) and reviews (5%). Most studies were conducted in The Netherlands, with the exception of four in other European countries and one in Southeast Asia. The studies either focused on individuals, professional use in primary care, hospital care and education, or collective use within local communities and for healthcare policymaking. Some of the studies concerned professional views on and experiences with the practical use and feasibility of the positive health method. Others focused on different target groups, ranging from children to elderly adults, and from patients with chronic diseases to people with low socio-economic statuses. See Table 1.

### 3.2. Themes and Sub-Themes

A total of five themes could be identified: implementation of PH and its impact (*n* = 13 articles), conceptual development (*n* = 16 articles), development of the PH method (*n* = 13 articles), measuring and monitoring effects (*n* = 5 articles), and scientific application (*n* = 25 articles). All of these themes and their sub-themes are discussed below.

#### 3.2.1. Implementation of PH and Its Impact 


*Primary care*


Within primary care, several projects are described in which the method of PH, using the dialogue tool and alternative dialogue, were applied as a practical method for implementing a holistic approach. PH aimed to improve connections between healthcare professionals and their patients and to foster a broad perspective on health during consultations [48,53,62,63]. Within two of these projects, PH was additionally used to serve as a shared framework through which organisations could implement changes to their organisational and financial structures in primary care [48,53]. These changes included testing a revised reimbursement system at the population level, initiating activities that foster a positive work environment, and promoting multidisciplinary collaboration. Furthermore, the PH dialogue tool was used in a healthy lifestyle programme for obese patients [58].

A quantitative analysis showed that the implementation of PH in primary care led to a 25% reduction in hospital referrals, as well as associated reductions in healthcare costs, improved quality of care, and greater job satisfaction [48]. Furthermore, a qualitative case study amongst GPs indicated that embracing PH empowers them to enhance and give substance to their holistic approach to patient care. It provided them with values, while also allowing them to take into account the broader context of the patient’s circumstances and desires. Furthermore, the study revealed that the incorporation of PH is related to changes in organisational conditions that would allow GPs to invest more time into their patients as well as into their own well-being. Overall, this led to a greater sense of job satisfaction [52]. Regarding the lifestyle programme for obese patients, positive effects over time were found for physical fitness, BMI, motivation, and (positive) health, with the largest effects being related to changes in the PH dimensions [58].


*Hospital care*


One article describes the use of the dialogue tool in two outpatient hospital departments as part of a research project. This qualitative study mentions several benefits reported by medical professionals, including a better understanding of individual patients’ circumstances and functioning, changed dynamics in resident–patient communication, and increased awareness of the value of patient-related outcomes and healthcare costs [24]. 

Another article highlights the application of PH at the management level in various hospitals. The concept of PH was reported to provide guidance for a changed strategic vision towards achieving a broader, patient-centred, integrated care approach [1]. 


*Education*


In an educational setting, the PH dialogue tool was incorporated into a module aimed at increasing health sciences’ and medical students’ understanding of the person behind the patient. The participating students could opt to use the tool to structure their conversations with patients dealing with chronic diseases. Students reported that these conversations increased their level of empathy and provided a more comprehensive understanding of patients’ experiences [61]. 


*Communities*


Studies conducted in community settings have found that PH is primarily used as a shared framework to facilitate joint efforts by citizens and local professionals in disadvantaged neighbourhoods to initiate health-promoting initiatives in each of the dimensions of PH [35,56,73]. The findings of a qualitative analysis revealed that the use of PH effectively facilitated the achievement of these objectives, catalysing an ongoing trend of transformation within neighbourhoods. PH has been found to enable citizens to embrace a new mindset that is characterised by more positive and adaptive thinking. For local professionals, PH enabled them to have a broader outlook on health, identify intrinsic needs, and empower citizens. In addition, local professionals were described as experiencing greater awareness of each professional’s unique role in supporting residents with regard to their health and facilitating community collaboration. Overall, PH integration seems to be more effective among local professionals than among citizens [73]. 


*National health and welfare policies*


A discourse analysis by the Dutch institute for Transitions shows that PH has acquired a prominent position in current Dutch healthcare and public health policy. The following development mechanisms for transition were applied: growing, strengthening, replicating, partnering, instrumentalising, and embedding. The authors indicate several main reasons for this success. They suggest that PH offers a unifying language that reflects the government trend of shifting from a disease-oriented perspective to one that is health-oriented. Providing a shared framework through the common language is also highlighted as one of the strengths of the concept, allowing it to function as an intermediary between the healthcare and social care sectors. Furthermore, the co-creation process involving different stakeholders during the development of the concept is identified as a contributing factor to its successful implementation [15]. 

#### 3.2.2. Conceptual Development

From the articles included in the rapid review, several recommendations emerged for the further development of PH as a conceptual model. These are summarised below. 


*Construct definition*


Some authors note that there may not be a single definition of health, and they emphasise that this should be taken into account when further elaborating the concept of PH. According to them, the conceptualisation of health depends on contextual factors and scientific perspectives [9,39,66]. They also emphasise that people construct their own understandings of health based on personal circumstances and individual characteristics [9,27,32,39,55,64,66]. For example, these articles highlight that people from lower socio-economic backgrounds have a more disease-oriented approach to health and focus on different aspects of health compared to other groups. To acknowledge viewpoints and interpretations of health, the formulation of the dynamic concept of health is considered a ‘general concept’ instead of a ‘definition’—based on sociologist Blumer’s ideas—to characterise rather than demarcate [44]. Furthermore, to include different viewpoints on health, the original study, which aimed to elaborate the dynamic concept and from which PH emerged, was based on a representative sample for the Dutch population [44,45]. In terms of content validity, an overlap between aspects within and between dimensions of the dialogue tool is noted by both users and experts [24,60]. For example, the differences between some aspects of the dimensions mental well-being, meaningfulness, and quality of life do not always seem clear. This overlap was confirmed by a factor analysis by Van Vliet et al. (2022) [14], who demonstrated that, although the six-dimensional structure seems appropriate to define the concept, adjustments to the arrangement of aspects of the dialogue tool are needed for the further refinement of PH as a construct. It is also mentioned that the current aspects of the dialogue tool include both elements that reflect health indicators and those that influence health determinants, such as housing conditions and one’s financial situation. Although these aspects are relevant in facilitating the ‘alternative dialogue’, they may not be appropriate to encompass a construct of PH [60]. Given the major influence of people’s living environments on their health, it is recommended to carefully consider incorporating this more explicitly into the concept [27,73]. Furthermore, the authors advise that how PH relates to other concepts, such as quality of life and participation, should be clarified [60]. 


*Philosophical and ethical implications*


Several philosophical comments were made. First, some researchers argued that focusing on PH and its relation to well-being and happiness could significantly push the boundaries of healthcare and potentially lead to further medicalisation [15,51,57]. Huber underlined that a broad perspective should instead promote collaboration between various domains, rather than increase the burden and responsibility of the medical sector [44]. An implementation study reported that this medicalising effect could not be observed in an experiment with PH in a hospital setting [24]. Second, several articles argued that resilience and self-management may not be attainable for everyone, as they require a certain level of strength that is not universally present [27,51,64]. Lastly, some concerns were raised about the notion of full accountability for one’s own health, which may mix up health and behaviour [57]. In reply, Huber (2016) refuted this by claiming that the focus is on strengthening ability and resilience, which should not be confused with behaviour or decisions about whether or not to take certain actions [44]. In addition, Huber emphasised that competence should be viewed in terms of people’s own capabilities. Guidance through the ‘alternative dialogue’ must be consistent with this.

#### 3.2.3. Development of the PH Method

As the number of professionals working with PH is increasing, several suggestions are made in the included literature for refinement and elaboration of the PH method.


*Applicability and inclusivity*


Various versions of the PH dialogue tool, the first version being for adults, are described in the included literature. These are intended for specific populations, such as children, adolescents, and people with lower reading skills [14]. Currently, the only development process published in a scientific journal is that of the children’s version [25]. Several evaluation studies recommended further elaboration of the PH method to enhance its applicability and effectiveness in various contexts and populations [24,53,73]. First, more clarity regarding the target population and setting is suggested [24]. One evaluation study in a hospital setting reported doctors mentioning the broad scope of dimensions that they felt went beyond the scope of some specialised medical professionals. According to them, the dialogue tool was mainly useful for patients with chronic and more complex conditions. Second, other concerns raised were about the current versions of the dialogue tool perhaps having the tendency to predominantly reflect a Western, individualistic perspective. Therefore, they recommended paying attention to the cultural sensitivity of the PH concept and method when applying it to non-Western populations [73]. Likewise, some studies advised that attention should be paid to aligning the PH method to suit vulnerable groups—including disadvantaged individuals and the sometimes frail elderly—as the literature indicates they may have a lower awareness of how their behaviour can influence their health and they may put a greater emphasis on illness than on health aspects [27,32,55,64]. Finally, a primary care evaluation study reported physicians noting the prevalence of interpretations of PH among professionals. According to them, this offers the opportunity for customisation and to align the implementation of PH as a shared framework suiting the preferences of each organisation, yet simultaneously introduces ambiguity regarding the optimal approach for its implementation [53].


*Suitability*


Perceptions of the PH dimensions were investigated in several studies to explore the suitability of the implementation of PH in practice [49,50,68,70]. A survey study confirmed that, as perceived by citizens, the six dimensions of PH capture health more effectively compared to the WHO domains (i.e., physical, mental, and social). Furthermore, it appears that youth health professionals (who work in the public healthcare domain) are generally positive about the concept. However, they indicate that, in their daily practice, there still seems to be a lack of focus on mental well-being and meaningfulness. Time constraints and the availability of numerous different conversation methods are reported by them to be barriers for implementation in practice [70]. A quantitative survey held among mental health care professionals also reported recognition of the importance of all dimensions of PH, except for daily functioning. Patients rated this dimension significantly higher compared to the professionals [68]. 

Although Dutch and Belgian nursing students acknowledged the significance of all PH dimensions, they expressed that quality of life and meaningfulness were comparatively less integrated into their curricula [49]. On the other hand, compared to patients, paramedic healthcare students regarded participation, mental well-being, and daily functioning as less important. Compared to patients, their lecturers rated the aspects of bodily functions and daily functioning as less important [50].

#### 3.2.4. Measuring and Monitoring the Effects of PH 

Several articles emphasised that the spiderweb is meant to serve as a dialogue tool rather than a measurement instrument [14,18,30]. However, Nahar (2022) stated that, along with the growing utilisation of PH, the need for measuring the impact of working with PH has increased [18]. To respond to this need, Van Vliet et al. (2021) have developed a 17-item measurement instrument through a psychometric study [14]. The items of this instrument were extracted from the aspects of the dialogue tool. The 17-item PH measurement instrument has been confirmed to have adequate psychometric properties in terms of its content and its convergent validity [30]. In order to obtain more insight into its measurement properties, the authors recommend further testing of the instrument, for example regarding its responsiveness, over time. In addition, several fundamental steps for the development of a measurement instrument are recommended, such as identification of the target population and refinement of the purpose of measurement [60]. In general, Nahar (2022) advises that, given the multidimensional, person-specific, context-dependent nature of health, the preferred methodological strategy for measuring effects would involve a mixed-methods approach that combines both quantitative and qualitative analyses [18].

#### 3.2.5. Scientific Application of PH


*Utilisation as a research method*


The PH concept and method appear to be extensively applied as both a quantitative and qualitative research method across a variety of scientific studies. This encompasses evaluations of lifestyle programmes (including e-coaching), arts therapy programmes, social interventions, and epidemiological studies. 

Despite the cautionary note that the dialogue tool is not a validated measurement instrument, 10 studies reported the use of the PH aspects to assess baseline characteristics or outcomes in healthcare-related interventions and cross-sectional studies. Among those, five studies utilised (or aimed to use in case of a study protocol) the 42 items (version 1.0) of the dialogue tool to calculate scores per dimension [26,35,36,65,72]. Others developed their own measurement systems [28,29,46,47,54]. In addition, one study mentioned the future use of a validated PH measurement instrument once it becomes available [31]. A study evaluating a holistic lifestyle intervention for obese patients employed the 17-item measurement instrument developed by Van Vliet et al. (2021). The study found more positive changes in outcome for the PH dimensions compared to traditional outcome parameters, such as physical fitness and BMI [14,58]. On the other hand, no changes for PH dimensions were found over time in a digital coaching programme for elderly people posing one question per dimension [47]. One cross-sectional study among children with severe chronic and/or congenital skin disorders reported relatively high scores on the PH dimensions, from which the authors concluded that these children were well able to adapt and self-manage [26]. 

Furthermore, eight articles reported that the dimensions of PH were part of their qualitative research methods, either during data collection [34,37,52,59,71] or as a framework during data analysis to organise themes based on the various dimensions [43,52,67,69,71].


*Input for developing related health concepts*


PH and its dimensions and aspects are utilised as an inspirational source to improve the International Classification of Functioning, Disability, and Health (ICF) [40,41]. When considering the dimensions of PH, studies propose a modified classification of the ICF domains. Furthermore, validation studies use PH aspects to assess the concurrent validity of a well-being scale [38] and also use it as a framework for evaluating the face validity of a quality-of-life measurement scale [42].

## 4. Discussion

### 4.1. Summary of Main Findings

This rapid review shows that, since the launch of PH in 2016, numerous scientific articles have emerged discussing the implementation of PH and its impact, its conceptual development, the development of the PH method, measurement and monitoring effects, and its scientific application. These articles highlight the transformative potential of PH in shifting from a disease-focused paradigm of healthcare to one that is health-oriented. Initial evaluation studies reveal positive results at both the individual and collective levels. However, several articles express the need for further refinement of the conceptualisation of PH and for ways to make the effects of PH more measurable. Professionals also express a desire for a more informed application and elaboration of the PH method across various settings and populations to enhance its effectiveness in practice.

### 4.2. Reflections

Scientific evaluation of effects is considered by some authors as an important prerequisite for the PH concept and method [14,15,18]. So far, this is mainly available from studies on primary and hospital care, as well as in local communities. In primary care, studies report a 25% reduction in hospital referrals and enhanced holistic patient care. Recent follow-up research confirms this reduction in referrals, also resulting in a decrease in total healthcare expenditure [74]. This approach has also been found to improve job satisfaction among primary care physicians. A holistic lifestyle programme for obese patients in which the dialogue tool was used showed favourable changes in positive health, physical fitness, and BMI [58]. In outpatient hospital departments, PH’s dialogue tool has been found to improve communication, patient awareness, and cost considerations. PH’s application on a hospital management level provides guidance for a changed strategic vision towards a more broad, patient-centred, and integrated care approach [1]. On a community level, PH serves as a shared framework for health promotion initiatives in disadvantaged neighbourhoods, although the tool’s specific role in achieving these goals requires clarification. On a national level, PH is gaining prominence in Dutch healthcare and public health policies [14,15]. Its language and unifying identity align with a shift from disease-oriented healthcare policy to health-oriented healthcare policy. Continued research is needed to further explore how and to which extent PH contributes to human health, quality of healthcare, and its impact on healthcare systems. This is still ongoing. A very recent study investigated, for example, which skills would be required to empower individuals in maintaining and improving their (positive) health, with a special focus on people with lower health skills and competences [75].

PH regards health as a multi-dimensional construct involving a wide range of life domains and disciplines. Its broad perspective aligns with the current trend in which collaborations across multiple disciplines and sectors to enhance overall health and well-being are stimulated [15,76,77]. It also aligns with the notion that the influence of the living environment on perceived health is undeniably significant [78]. Recently, the institute for Positive Health (iPH) drafted a knowledge agenda, aiming to provide direction for research and the development of PH in the coming years. In consultation with citizens, policymakers, researchers, health insurance companies, and healthcare professionals, knowledge questions were clustered into the following domains: essence of PH, human, living and living environment, education, work, health and well-being, and ecosystem (see Figure 3). As PH is a concept that is originally rooted in healthcare, it is not surprising that this review reveals that PH-related studies still mainly focus on the field of healthcare. However, in line with its multi-dimensional foundation and societal trends, more research on broader domains as stipulated by the PH research agenda is warranted. This may enhance our understanding of effective and ineffective approaches in the various domains, thereby contributing to the knowledge of where and how to implement the PH concept and method.

Initially, due to the substantial interest from healthcare professionals when PH was first introduced, the main focus among researchers was on empirical testing of PH within the healthcare practice before delving deeper into its scientific framework. This enabled co-creation with pioneering professionals in the field to iteratively refine the method. The results, as shown by this review, demonstrate that, even though we may not possess a comprehensive understanding of the underlying mechanisms, the effects appear promising. However, to enable a sustainable integration of the concept, it is also important to address scientific questions that have been raised regarding the conceptualisation of PH. These include suggestions to formulate a more elaborated description of the concept of PH, the development of a PH construct with a corresponding measurement scale, the relationship with other concepts (such as well-being and quality of life), and a greater focus on the living environment. Regarding the PH method, suggestions have been made with respect to more clarity on suitable settings and groups, more attention for the inclusivity of vulnerable groups, and more specific guidelines for implementation of PH in practice. To address this, a handbook for use of PH in primary care was recently written [10]. The questions and suggestions presented in this rapid review may serve as a valuable starting point for further consideration and prioritisation of the steps deemed necessary for the continued development and refinement of the PH concept and method. In this regard, an ongoing dialogue between academia and practice may play a pivotal role in facilitating a sustainable implementation of PH across various domains and organisational levels. For example, through dialogue between practitioners and researchers, the feasibility and desirability of refining the PH methodology can be explored and assessed to suit various settings and populations. This may provide a clear foundation for professionals working with PH, while allowing sufficient room for flexibility and aligning with the specific context of each practice and setting. In this regard, descriptions of the core elements of the PH methodology are studied in an ongoing project. Clarity regarding these core elements is important, as this may guide professionals in practice to adequately implement the concept and method. In addition, complete and accurate information is needed in order to avoid misunderstandings among individuals completing the PH tool. This uniformity in interpretation and adequate use is also regarded as a prerequisite for scientific research on the effectiveness of PH [79]. Moreover, as not everyone may have the same level of access to technology and healthcare resources, attention should also be paid to the ways in which PH could be promoted among those demographics. Furthermore, regarding the wish for a clearer definition and demarcation of the PH concept, as appears from certain studies, it is essential to determine in dialogue with stakeholders the extent to which this is desirable, considering PH’s elaboration from a ‘general concept’ rather than a rigid definition. From an international perspective, various interpretations of positive health have emerged, all rooted in a salutogenic approach. In this context, researchers are aiming to achieve greater consistency and clarity with regard to this salutogenic approach [9]. Research on PH may contribute to this movement, for instance, by further delineating guiding principles, particularly focused on aspects such as resilience, meaningfulness, self-care, and patient empowerment. In addition, this rapid review indicates that the majority of studies so far have been conducted in The Netherlands. This may be due to the proactive adoption of PH principles by the Dutch government. This fostered widespread support and recognition of PH, resulting in various implementations and research projects within the country. For the future, it would be valuable to initiate more international research in order to gain insights into the effects of PH in various cultural contexts. These efforts are vital given the pressing need to address global health challenges and the demands on healthcare systems [2,3,4].

From this rapid review, it appears that researchers are using a variety of quantitative, qualitative, and mixed-methods approaches to evaluate PH interventions. Given the person-centred, bottom-up, and complex systemic perspectives that form the core of PH interventions, it is important to collectively explore scientific approaches that align with these elements. For impact studies, previous recommendations have emphasised the importance of adopting a mixed-methods approach incorporating both qualitative and quantitative methods [18]. Further insights can be used from methodologies such as the Developmental Evaluation approach and action research, which place a stronger emphasis on fostering innovation or change within a complex adaptive system, rather than adhering to a rigid pre-defined evaluation protocol [80,81]. The ENCOMPASS framework serves as a good example of such guidance [82]. In addition, the principles of ‘citizen science’ and ‘co-creation’, which entail active involvement of the target group throughout the research project from design to execution to completion, appear to align well [83]. Furthermore, attention should be paid to inclusivity during research, with the aim of making all aspects of the research accessible and understandable for all involved stakeholders [84]. Finally, in recognition of the need for a suitable measurement instrument to assess changes in line with PH, the 17-item measurement instrument was developed to fill this gap [14,30]. This instrument is currently the subject of an ongoing project for further testing and optimisation to enhance its quality for application in scientific evaluations. The study by Philippens et al. (2021), which found greater positive changes in PH dimensions as measured by the 17-item instrument compared to more traditional parameters, such as BMI and physical fitness, shows the importance of a broad, holistic perspective beyond physical measurements in lifestyle programmes [58]. However, given the widespread use of the PH dialogue tool as a measurement instrument in various evaluation studies, it is important to clearly convey that the dialogue tool itself is not recommended for this purpose.

In summary, this rapid review proposes several recommendations for future research on PH, which are discussed above and summarized in Box 1. It is encouraging that ZonMw, the Dutch national funding body for health research and development, has continued to incorporate PH into their research programmes and objectives [85]. This continued research can ultimately contribute to the advancement of healthcare transformation towards more health-oriented care.

Box 1Recommendations for future research.
➢Initiation of more research on effects of PH on broader domains than health and social care, such as education, work, the living environment, and eco-systems.➢Prioritisation of suggested research topics that should be addressed to refine the conceptualisation of PH. This includes suggestions to formulate a more elaborated description of PH, development of a PH construct with a corresponding measurement scale, relationship with other concepts (e.g., well-being, quality of life), and a greater focus on the living environment.➢More insight into the core elements or guiding principles of PH, which may guide practitioners to implement PH effectively in various contexts.➢More clarity regarding suitable settings and target groups for the implementation of PH, with special attention paid to inclusivity of vulnerable populations.➢Continued research to further explore how and to which extent PH contributes to human health, quality of healthcare, and its impact on healthcare systems. ➢Stimulation of international research collaborations to gain insight into the effects of PH in various cultural contexts.➢Ongoing dialogue between academia and practice to facilitate the sustainable integration of PH across various domains and organisational levels. ➢Further exploration of scientific approaches that align with the person-centred, bottom-up, and complex system perspectives inherent to PH interventions, and that are able to capture the multifaceted nature of PH and its impact.


### 4.3. Methodological Considerations

One of the core strengths of this study format lies in its ability to compile all available scientific research within a relatively short timeframe. However, while we aimed to conduct this process as systematically and meticulously as possible, it is possible that we may have missed scientific studies on PH published in other databases. To ensure the quality of the included publications, we chose to exclusively consider studies published in peer-reviewed scientific journals. This decision led us to exclude other reports and grey literature, which may have limited our insight into developments that were reported directly from policy and practice. Another limitation is that we did not assess the quality of each study individually, and as a result, we cannot make statements about the accuracy of the reported results. Lastly, the initiative to conduct this rapid review originated from researchers at iPH and PHi, some of whom are also authors of certain publications which we included. To minimise the risk of bias, two researchers from academic institutions were engaged as co-authors in this study, with the task of reviewing both the methodology and interim results at various stages throughout the research process.

## 5. Conclusions

In summary, this rapid review highlights both the potential of PH as a transformative innovation and the challenges for the further development of the concept and its method. It underscores the need for continued research, interdisciplinary and international collaboration, and methodological innovation to achieve the full potential of PH in improving health, enhancing healthcare quality, and reshaping public health and healthcare systems as a whole. Based on the recommendations in this rapid review, we can move closer to these goals.

## Figures and Tables

**Figure 1 healthcare-12-00671-f001:**
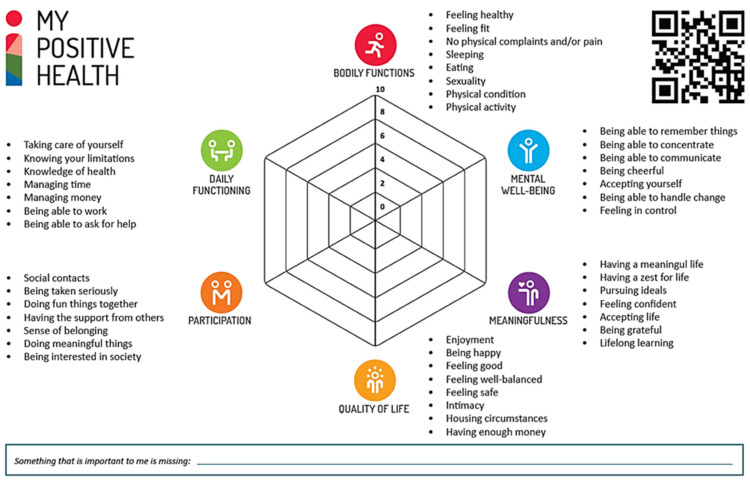
My Positive Health dialogue tool for adults. Retrieved from: https://www.iph.nl/en/participate/free-downloads/ (accessed on 4 March 2024). Reproduced with permission by the Institute for Positive Health, The Netherlands, 2024.

**Figure 2 healthcare-12-00671-f002:**
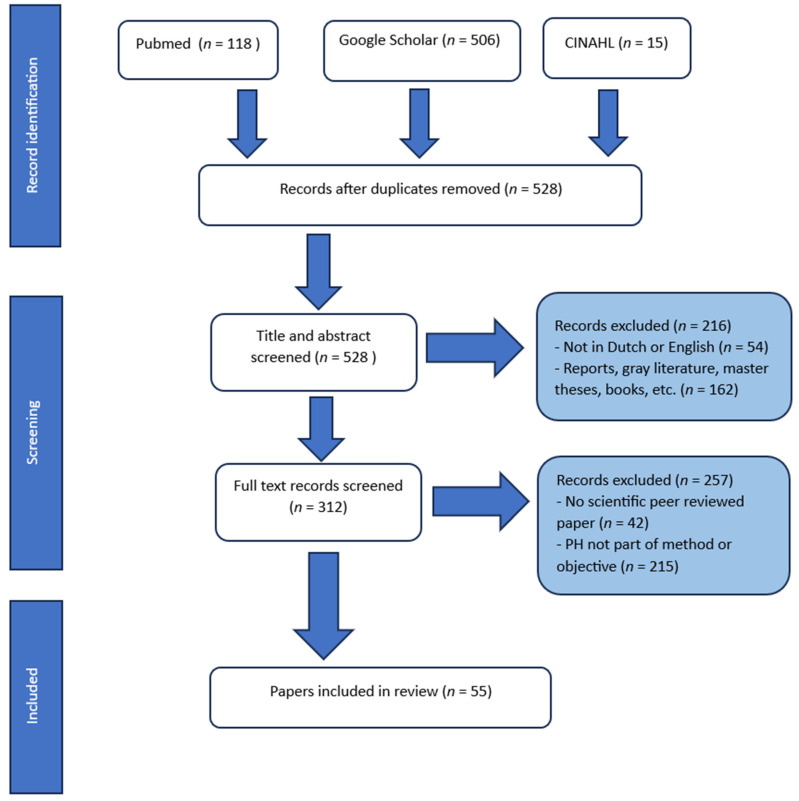
PRISMA flow diagram for identification, screening, and inclusion of papers.

**Figure 3 healthcare-12-00671-f003:**
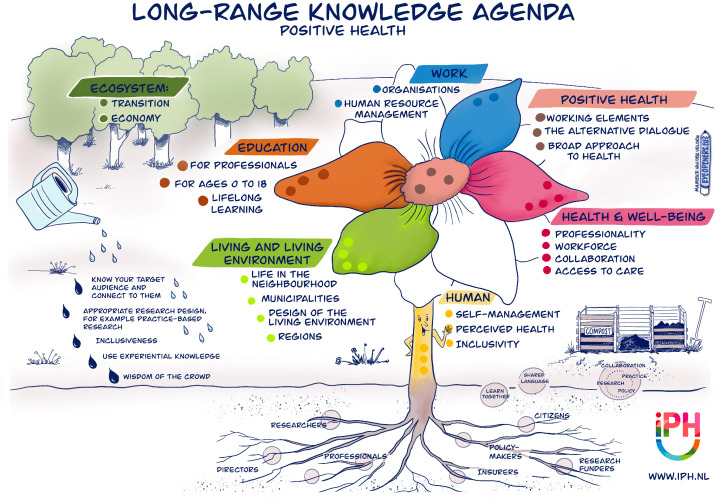
Long-range knowledge agenda for Positive Health by Marieke van der Velden-visual storyteller-Eyeopeners. Retrieved from https://www.zonmw.nl/nl/nieuws/kennisagenda-positieve-gezondheid-aangeboden-aan-zonmw (accessed on 4 March 2024). Reproduced with permission by the Institute for Positive Health, The Netherlands, 2024.

**Table 1 healthcare-12-00671-t001:** Overview of the articles included in the rapid review.

Authors	Language	Design	Study Population	Setting *
Bock et al. (2021) [24]	English	qualitative	medical specialists	hospital
Bodryzlova et al. (2023) [9]	English	review	n/a	research
De Jong-Witjes et al. (2022) [25]	English	qualitative	children	hospital and community care
De Maeseneer et al. (2019) [26]	English	mixed-methods	children	hospital (Belgium)
Den Broeder et al. (2017) [27]	English	qualitative	health and welfare professionals	community
Dierx & Kasper (2022) [28]	English	quantitative	citizens	research
Dierx & Kasper (2018) [29]	Dutch	quantitative	citizens	research
Doornenbal et al. (2022) [30]	English	quantitative	citizens	research
Elsner et al. (2022) [31]	English	mixed-methods	citizens	community (Europe)
Flinterman et al. (2019) [32]	Dutch	qualitative	citizens with low SES	community
Flinterman et al. (2019) [33]	Dutch	review	n/a	public healthcare and social work
Groot et al. (2021) [34]	English	qualitative	elderly adults	community and care
Grootjans et al. (2019) [35]	English	mixed-methods	citizens with low SES	community
Haaps et al. (2023) [36]	English	quantitative	patients	community
Hamm-Faber et al. (2020) [37]	English	qualitative	patients	hospital
Haspels et al. (2023) [38]	English	quantitative	citizens	research
Haverkamp et al. (2017) [39]	Dutch	reflection	citizens with low SES	community
Heerkens et al. (2017) [40]	Dutch	mixed-methods	n/a	healthcare; research
Heerkens et al. (2018) [41]	English	mixed-methods	n/a	healthcare; research
Hendrikx et al. (2019) [42]	English	quantitative	citizens	research
Huang et al. (2020) [43]	English	qualitative	elderly adults	hospital (Belgium)
Huber et al. (2016) [44]	Dutch	mixed-methods	stakeholders in healthcare	healthcare professionals
Huber (2016) [45]	Dutch	reflection	n/a	research
Hurmuz et al. (2022) [46]	English	mixed-methods	elderly adults	community
Hurmuz et al. (2020) [47]	English	mixed-methods	elderly adults	community
Johansen (2018) [1]	English	qualitative	healthcare organisations	care and cure
Johansen et al. (2023) [15]	Dutch	qualitative	n/a	national healthcare system
Jung et al. (2018) [48]	Dutch	quantitative	patients	primary care
Kablau, Kiki et al. (2020) [49]	Dutch	qualitative	bachelor students	education (Belgium, The Netherlands)
Karel et al. (2019) [50]	English	quantitative	bachelor students and lecturers	education
Kingma (2017) [51]	Dutch	reflection	n/a	research
Kramer et al. (2021) [52]	English	qualitative	elderly adults	community
Lemmen et al. (2021) [53]	English	qualitative	primary care professionals	primary care
Moens et al. (2022) [54]	English	quantitative	elderly adults	community
Nahar et al. (2022) [18]	Dutch	reflection	n/a	research
Platzer et al. (2021) [55]	English	qualitative	elderly adults with low SES	community
Pardoel et al. (2022) [56]	English	qualitative	citizens	community (Southeast Asia)
Poiesz (2016) [57]	Dutch	reflection	n/a	research
Philippens et al. (2021) [58]	English	quantitative	overweight and obese patients	primary care
Prevo et al. (2020) [59]	English	mixed-methods	low-SES community	community
Prinsen & Terwee (2019) [60]	English	qualitative	stakeholders in healthcare	research
Romme et al. (2020) [61]	English	qualitative	healthcare students	education
Smeets et al. (2021) [62]	English	qualitative	patients with chronic conditions	primary care
Smeets et al. (2022) [63]	English	qualitative	patients with chronic conditions	primary care
Stronks et al. (2018) [64]	English	qualitative	citizens	community
Van Doorn et al. (2021) [65]	English	mixed-methods	adolescents with mental problems	mental healthcare
Van Druten et al. (2022) [66]	English	review	n/a	research
Van Everdingen et al. (2021) [67]	English	mixed-methods	homeless people	community
Van de Loo et al. (2022) [68]	Dutch	quantitative	mental healthcare professionals	mental healthcare
Van Lonkhuizen et al. (2021) [69]	English	qualitative	patients with Huntington’s disease	community
Van Meerten et al. (2020) [70]	Dutch	mixed-methods	youth health professionals	community
Van Sleeuwen et al. (2020) [71]	English	qualitative	caregivers of ICU patients	hospital
Van Velsen et al. (2019) [72]	English	quantitative	elderly adults	community and care
Van Vliet et al. (2021) [14]	English	quantitative	n/a	research
Van Wietmarschen et al. (2022) [73]	English	qualitative	citizens with low SES	community

n/a = not applicable; SES = socio-economic status; * Dutch setting if not specified.

## Data Availability

Reasonable requests for sharing data can be made by sending an email to the corresponding author.

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
