# Peer review of "Rapid Review on the Concept of Positive Health and Its Implementation in Practice"

_healthcare, 2024, doi:10.3390/healthcare12060671_

Round 1
Reviewer 1 Report
Comments and Suggestions for Authors
The article presents the results of a rapid review, highlighting the transformative potential of Positive Health (PH) in shifting from a disease-focused to a health-oriented paradigm.
Introduction
In its introduction section, the paper provides a clear overview on the concepts of PH and related tools and procedures, and defines the aims of the rapid review.
However, it remains unclear if and how the concept of PH relates to existing concepts of health (e.g. WHO), health promotion, quality of life, as well as the social determinants for health which partly seem to be used as context factors (e.g. housing).
Materials and Methods
A rapid review approach following the Cochrane Rapid Re- 103 view recommendations was applied. The review steps, guiding questions, literature search strategy and inclusion criteria are presented in a well-structured way.
Results
In general, the result section is clear and easy to follow.
In the key themes section it would be good to have numbers included, e.g. the number of articles identified per theme.
For health care in the primary care setting and in the hospital setting, the description could profit from a separated presentation of impact on level of individuals and on level of organisation.
Line 197: it remains unclear why the heading is “hospital and care”. In the table above care seems to be closely related to community, and the paragraph under this headline is only focusing on hospitals. Please consider if the heading should be just “hospital” or “hospital care”.
Reflections:
Line 402f: “PH’s application at the management level of hospitals offered an opportunity to a more sustainable healthcare vision”. Please elaborate for better understanding.
Line 411: “skills to maintain and improve their PH”. Please reformulate as this part of the sentence doesn’t relate to the previous sentence.
Conclusions
Conclusions are supported by the results and reflections.
However, please rethink the stated potential of PH to reshape “society as a whole”.
Text editing;
Line 77, 80: please check punctuation (commas)
Line 148: clearer: regular meetings of the group of authors
Line 227: please check [wietm]
Line 375: please check [hearkens, heerkens]
Author Response
Dear reviewer,
We thank you for the positive feedback on our manuscript and the clear and thorough comments. We have considered all remarks and have revised the manuscript accordingly. Please see below:
- It remains unclear if and how the concept of PH relates to existing concepts of health (e.g. WHO), health promotion, quality of life, as well as the social determinants for health which partly seem to be used as context factors (e.g. housing).
PH has emerged as a more broad and dynamic perspective on health in addition to the traditional definition of the WHO. Thereby it overlaps with concepts as wellbeing, quality of life, etc. As mentioned by one of the included papers in the review (line 306): how exactly is not yet studied in detail and is one of the topics that is advised by researchers in the included papers to be addressed. We have added a sentence in the introduction to mentioned this (line 33-55). Same about your remark regarding the distinction between aspects and determinants of health. This is discussed in the result section (line 299-305).
- In the key themes section it would be good to have numbers included, e.g. the number of articles identified per theme.
We have added the number of articles per theme.
- Line 197: it remains unclear why the heading is “hospital and care”. In the table above care seems to be closely related to community, and the paragraph under this headline is only focusing on hospitals. Please consider if the heading should be just “hospital” or “hospital care”.
Thank you for your close look. We have changed the heading in ‘hospital care’.
- Line 402f: “PH’s application at the management level of hospitals offered an opportunity to a more sustainable healthcare vision”. Please elaborate for better understanding.
We rephrased the sentence: PH’s application at the management level of hospitals provided guidance for a changed strategic vision towards a more broad, patient-centered and integrated care approach (line 451-454)
- Line 411: “skills to maintain and improve their PH”. Please reformulate as this part of the sentence doesn’t relate to the previous sentence.
We have reformulated this sentence (line 439).
- Conclusion: please rethink the stated potential of PH to reshape “society as a whole”.
We agree with you and changed this in public health and healthcare systems as a whole
- Line 77, 80: please check punctuation (commas)
Revised accordingly
- Line 148: clearer: regular meetings of the group of authors
In consultation with a native speaker changed in ‘in which all authors participated’ (line 183)
- Line 227: please check [wietm]
We have now inserted the correct reference number
10. Line 375: please check [hearkens, heerkens]
We have now inserted the correct reference numbers
Reviewer 2 Report
Comments and Suggestions for Authors
Review manuscript: 2854867
The authors present an interesting rapid review of the concept of Positive Health. I think it reads well and adds considerable information on addressing PH and its potential implications on the healthcare systems and corresponding challenges in developing such concepts and its methods. However, I think the paper needs revision to address the formatting and language issues.
Below are some of my specific comments:
- The English language needs to be refined.
- I believe that the tool's accuracy may be affected by the user's incomplete or inaccurate information during the self-assessment process, which may create a potential gap in understanding their health status. Could authors detail the effective functioning of such tools and or developmental methods in addressing such false positive and negative results?
- Similarly, not everyone may have the same level of access to technology and health care resources, and in what way PH could be promoted in those demographics?
- It looks like some of the references were not compiled appropriately, See lines 226 and 374.
The English language requires a moderate editing.
Author Response
Dear reviewer,
We thank you for the positive feedback on our manuscript and the clear and thorough comments. We have considered all remarks and have revised the manuscript accordingly. Please see below:
- The English language needs to be refined.
To improve language, the text is revised by an English native speaker.
- I believe that the tool's accuracy may be affected by the user's incomplete or inaccurate information during the self-assessment process, which may create a potential gap in understanding their health status. Could authors detail the effective functioning of such tools and or developmental methods in addressing such false positive and negative results?
We agree with you that sufficient understanding of the tool for optimal functioning is of utmost importance. Both for the individual who completes the questionnaire as well as for the professional who guides the conversation. And of course for researchers to be able to study its effectiveness. We address this issue now more explicitly in line 504-507.
- Similarly, not everyone may have the same level of access to technology and health care resources, and in what way PH could be promoted in those demographics?
This is a relevant remark about a topic that, although mentioning attention for vulnerable groups, hasn’t been specifically addressed yet. We therefore have included this remark in our recommendations for future studies. See line 510 – 512.
- It looks like some of the references were not compiled appropriately, See lines 226 and 374.
Thanks for your close look. We have inserted the correct reference numbers.
Reviewer 3 Report
Comments and Suggestions for Authors
Dear Authors,
Thank you for submitting your manuscript titled "Rapid Review on the Concept of Positive Health and its Implementation in Practice" for consideration.
Your literature review is thorough. However, for the manuscript to reach its full potential, I recommend clarifying more detailed the methodology used for literature selection and analysis.
In conclusion, your manuscript makes an important contribution to the field. With the suggested revision, I believe it will be a valuable resource for both researchers and practitioners interested in Positive Health.
Best regards,
Author Response
Dear reviewer,
We thank you for the positive feedback on our manuscript and your clear comments. We have considered your remark and have revised the manuscript accordingly. Please see below:
I recommend clarifying more detailed the methodology used for literature selection and analysis.
We made some extra clarifications in the text of the methodology section.
Reviewer 4 Report
Comments and Suggestions for Authors
The study looked at positive health and its implementation in practice. This was an interesting study which would help in giving optimum health to individuals and communities.
Line 147: The co-authors should mentioned at the back of the paper when indicating the work done by the different co-authors.
Lines 439-440: PH is confusing. Different people have their different explanations and definitions. Would we want to call the effects mentioned in this study the effects of PH if there was no comprehensive understanding of the underlying mechanisms?
I would suggest that consensus has to reached on the definition and explanation of PH before we start talking about its benefit and implementation. PH explanation is too broad, subjective and means something different to different people.
Author Response
Dear reviewer,
We thank you for the feedback on our manuscript and the clear and thorough comments. We have considered all remarks and have revised the manuscript accordingly. Please see below:
- 1. In the body of the manuscript I haven’t find the answers to the questions like. I would advise to explain above points in an introduction:
* For whom HP is it created?
* How it can influence health care system (or others)?
* Who is its main beneficiary?
* How HP can be implement in a health care system if it is (in many points) descriptive=subjective?
The method was initially meant for use by professionals in healthcare and social domain. Soon after within conversations between peers or volunteers and after that other domains followed sych as work an education. We have now explained this in line 58-62.
Furthermore, we included an explanation on how it can influence human health and wellbeing (its main beneficiary) and how it can change healthcare system (broad perspective, genuine attention, increased collaboration across various domains (i.e. integrated care)). Also about your remark regarding subjectivity: it provides a more holistic perspective in addition to a more traditional approach which is widely prevalent, comprising e.g. physical checks and medication prescriptions. See line 76-85.
- The authors should try to explain why (almost) only Dutch scientists (table 1 and cited literature) are interested in the issues discussed.
Thank you for this good point. We have inserted a reflection on this in the discussion (line 521 – 526)
- Conclusions should be more precisely, even if they are just guidelines, the recommendations cannot be so general.
We agree with you that the conclusion was quite general. However, since the discussion contains many recommendations we decided not to include them all in the conclusion. Instead, we created an extra box summarising all recommendations at the end of the discussion section. In the conclusion we refer to this box. See line 557 – 558, 564; line 586-587.
Reviewer 5 Report
Comments and Suggestions for Authors
Review articles have their own rules, which are difficult to verify; one can only rely on the reliability of the cited literature (exhaustive sources) and their objective assessment. The manuscript titled „Rapid review on the concept of Positive Health and its implementation in practice” presents quite interesting data. However authors by deciding to describe PH should introduce much more widely its positive health influence.
In the body of the manuscript I haven’t find the answers to the questions like:
- For whom HP is it created?
- How it can influence health care system (or others)?
- Who is its main beneficiary?
- How HP can be implement in a health care system if it is (in many points) descriptive=subjective?
I would advise to explain above points in an introduction.
The authors should try to explain why (almost) only Dutch scientists (table 1 and cited literature) are interested in the issues discussed.
Conclusions should be more precisely, even if they are just guidelines, the recommendations cannot be so general.
Author Response
Dear reviewer,
We thank you for the positive feedback on our manuscript and the clear and thorough comments. We have considered all remarks and have revised the manuscript accordingly. Please see below:
- Line 147: The co-authors should mentioned at the back of the paper when indicating the work done by the different co-authors.
Contribution of all co-authors is mentioned at the back of the paper according to the journal’s format. Line 589- 592.
- Lines 439-440: PH is confusing. Different people have their different explanations and definitions. Would we want to call the effects mentioned in this study the effects of PH if there was no comprehensive understanding of the underlying mechanisms?
We agree with you that this is an important topic to be addressed. We have added some text about the importance of improved uniformity and clarity regarding the principles of PH to guide professionals in the field and as a prerequisite to scientifically study the effectiveness of PH. This is actually a topic by and ongoing study by Maastricht University. See line 504 to 510.
- I would suggest that consensus has to reached on the definition and explanation of PH before we start talking about its benefit and implementation. PH explanation is too broad, subjective and means something different to different people.
Yes, this is indeed an important issue that requires attention. It sometimes appears to be a tension between Blumer's principles as a 'general concept' and the need for greater clarity and uniformity. We address this matter in lines 488 to 493, line 504 -510, and line 513-516 .